# A new program for systematically enhancing cognitive reserve in healthy adults: A pilot randomized active-controlled clinical trial

Carol Kotliar[1,2]*, Lisandro Olmos[1,3], Martín Koretzky[1], Ricardo Jauregui[1], Tomás Delía[1‡], Oscar Cingolani[4]

1 Espacio Santa María, Buenos Aires, Argentina, 2 Institute for Biomedical Research (BIOMED), School of Medical Sciences, Pontifical Catholic University of Argentina (UCA), National Scientific and Technical Research Council (CONICET), Buenos Aires, Argentina, 3 Universidad Barceló, Buenos Aires, Argentina, 4 Johns Hopkins Hospital, Baltimore, Maryland, Estados Unidos

☽ These authors contributed equally to this work.
‡ These authors also contributed equally to this work.
* ckotliar@gmail.com

## Abstract

### Objective

To evaluate the effectiveness of the Mental Training Tech 24.5 (MTT24.5) cognitive stimulation program, designed to enhance cognitive performance and neuroplasticity in healthy adults.

### Background

Cognitive decline is a significant concern in aging populations, with research suggesting that neuroplasticity and cognitive reserve can be enhanced through targeted cognitive training. The MTT24.5 program aims to stimulate brain function through a combination of new knowledge acquisition (DATA) and learning techniques (TECHS), organized into a systematic algorithm. This approach may offer a novel way to prevent or mitigate age-related cognitive decline.

### Design

Pilot clinical study, active-controlled, open randomization.

### Setting

Adults from the general population with no clinical cognitive deterioration, recruited from three sites within the Autonomous City of Buenos Aires and its metropolitan area.

**Data availability statement:** All relevant data are within the paper and its Supporting Information files.

**Funding:** The author(s) received no specific funding for this work.

**Competing interests:** The authors have declared that no competing interests exist.

## Participants

120 volunteers were enrolled, of which 76 participants (56 in the intervention group, 20 in the control group) met the study requirements and selected a site closest to their residence.

## Methods

The MTT24.5 program consists of 12 weekly in-person sessions (totaling 24.5 hours), during which participants learned 40 knowledge units (DATA) and 100 learning techniques (TECHS). These were organized into binomials, where each unit of DATA was paired with 3–4 TECHS. Pre- and post-intervention assessments included medical history, lifestyle factors, cognitive reserve scale, Addenbrooke's Cognitive Examination-Revised (ACE-R), and Mini-Mental State Examination (MMSE).

## Results

The mean age was 59 years for both groups. Baseline ACE-R scores were comparable (91.3). The global cognitive score increased by 4.6 points (5%) in the intervention group compared to a decrease of 0.5 points in the control group ($p < 0.001$). The most significant improvement was observed in the memory domain (2.4 points, 11.4% increase) versus a 0.3-point increase in the control group ($p < 0.007$), with secondary improvements in verbal fluency, language, and visuospatial skills. Notably, participants with baseline ACE-R scores below 85 showed greater improvements ($p < 0.003$). The effects were consistent across various phenotypic factors, such as age, sex, chronic disease distribution, and lifestyle.

## Conclusions

The MTT24.5 program, based on a systematic algorithm for acquiring new knowledge and skills, significantly enhances cognitive reserve and overall cognitive performance, particularly in individuals with lower baseline cognitive scores. These findings suggest that structured cognitive stimulation could play a critical role in preventing cognitive decline and promoting cognitive health in healthy adults. Given the promising results, future studies involving larger populations and long-term follow-up are essential to validate these effects and explore the potential for mitigating age-related cognitive decline and enhancing quality of life.

## Registration

The study was registered in accordance with local regulations at the National Council for Scientific and Technological Research (CONICET) – Institute of Biomedical Research (BIOMED), and also in the National Ethics Committee, and at clinicaltrials.gov (NCT06549517).

## Introduction

As global life expectancy continues to rise [1–3], the need for effective strategies to enhance cognitive reserve throughout the lifespan has become increasingly important. The key cognitive functions that sustain an autonomously functioning life include attention, memory, verbal fluency, language, and visuospatial skills [4–6]. Efforts to sustain and enhance them throughout life can promote neuroplasticity [7,8], thereby strengthening cognitive reserve [9], with factors such as age, experience, and environment modulating this adaptive brain process [10,11]. However, most interventions aimed at promoting cognitive function and neuroplasticity have primarily focused on older adults already experiencing cognitive decline [12–15]. A truly transformative approach would advocate for the integration of continuous brain training strategies throughout the lifespan, even in cognitively healthy adults without clinical signs of cognitive deterioration. By progressively and sustainably enhancing cognitive reserve through novel strategies, such an approach could potentially prevent, delay, or reduce age-related cognitive decline.

There have been novel studies such as Hardy et al [16] that sustain the benefits of interactive platforms such as Lumosity which aimed, and succeeded to, demonstrate that progressively challenging, targeted cognitive training can be an effective tool for improving core cognitive abilities including speed of processing, working memory, and fluid reasoning. The results there presented are a solid demonstration that cognitive training programs targeting a variety of cognitive capacities witch different exercises can be more effective than traditional activities such as crossword puzzles and Sudoku at improving a broad range of cognitive abilities.

The Nun Study, led by epidemiologist David Snowdon, investigated the cognitive health of 678 nuns who remained intellectually active throughout their lives [17]. They were regularly engaged in studying, teaching, and other cognitively demanding activities, with their cognitive test scores remaining normal and stable until their deaths. Postmortem examinations of their brains revealed significant neuropathological markers consistent with Alzheimer's disease, including neurofibrillary tangles and amyloid plaques [18]. Despite the presence of these pathological changes, none of the participants exhibited clinical signs of dementia. These findings suggest that lifelong intellectual engagement contributed to the development of a substantial cognitive reserve, which may have played a protective role, allowing the brain to compensate for the neuropathological burden of Alzheimer's disease and thereby preserve cognitive function. This underscores the potential of continuous cognitive stimulation in mitigating the clinical manifestations of neurodegeneration.[19] Similarly, research on London taxi drivers has shown that extensive spatial navigation training leads to structural brain changes [20]. These drivers undergo rigorous training, which involves memorizing the names and layouts of over 25,000 streets and thousands of points of interest in London. Functional magnetic resonance imaging (fMRI) studies have revealed that this intensive spatial navigation expertise is associated with increased gray matter volume in the posterior hippocampus, a brain region crucial for memory and spatial navigation [19]. Furthermore, the amount of navigation experience correlates with hippocampal gray matter volume, suggesting that spatial knowledge and experience are directly linked to structural brain changes [20,21]. Lessons from the Nun and Taxi Driver studies demonstrate that interventions introducing novel learning experiences and the acquisition of new skills provide a more robust stimulus for neuroplasticity. In contrast, many existing cognitive stimulation programs focus on re-training skills already acquired, often through the reinforcement of familiar knowledge and memory, which may offer less stimulation for brain adaptation and plasticity [22–25].

Our new program, Mental Training Tech 24.5 (MTT24.5), is based on a structured system of brief learning sessions encompassing the four basic sciences, with no interrelated connections between them, aiming to stimulate cognitive reserve through diverse brain areas. This approach incorporates memory techniques, including the use of codes and braille literacy [26,27], as well as activities targeting the non-dominant hemisphere of the brain [28]. With a panel of 100 distinct tasks, the program is designed to challenge participants with new and unfamiliar material, thereby promoting neuroplastic changes.

Notably, the study focuses on cognitively healthy adults without clinical signs of cognitive impairment, distinguishing it from previous studies that have primarily involved individuals with cognitive decline [12–15]. This clinical trial represents

the general population, with typical rates of chronic non-communicable diseases [29], offering a valuable opportunity to evaluate the broader applicability of this cognitive training methodology.

Through this pilot design, we aim to develop a hypothesis to validate the effectiveness of the MTT24.5 's cognitive learning interventions for improving cognitive performance in healthy adults. Future studies could expand on this work by correlating the effects with long-term follow-up throughout the lifespan, with the goal of analyzing its potential impact on the reduction or delay of age-associated cognitive decline risk.

## Objectives

Primary objective: to assess the cognitive effects of MTT24.5 in global cognitive performance among healthy adults. Secondary objective: to evaluate effects of MTT24.5 in cognitive domains of memory, attention, verbal fluency, language, and visuospatial skills in these population.

## Methods

### Design

This is a randomized, controlled, open-label trial involving 76 healthy middle-aged adults. Participants were randomly assigned in a 2:1 allocation ratio to two groups: an intervention group that completed the cognitive stimulation program (n = 56) and a control group (n = 20) (Fig 1).

Participants attended 12 in-person sessions of the MTT24.5 cognitive stimulation program and were required to meet the following

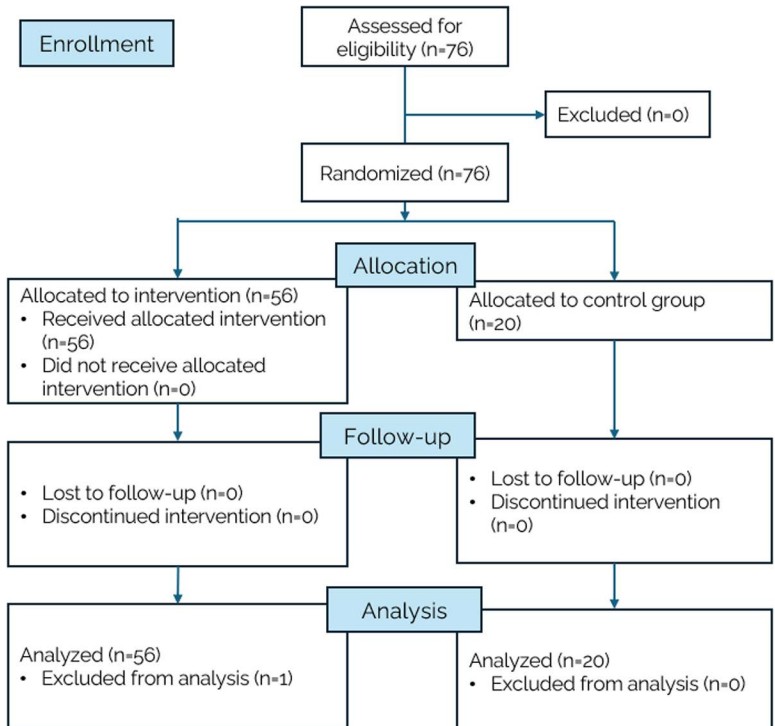

**Fig 1. CONSORT flow diagram of participant recruitment and retention.** This diagram illustrates the flow of participants throughout the study, from initial screening to final analysis. A total of 56 participants were randomly assigned to the MTT24.5 cognitive training program (intervention group), and 20 were assigned to the control group.

1] *inclusion criteria:* both sexes, aged 45–80 years, physical and mental autonomy for daily living; and

2] *exclusion criteria:* no cognitive complaints or clinical manifestations of cognitive impairment, and no impairments in writing, hearing, or reading that would interfere with participation. Additionally, participants could not be taking medications affecting cognition or be enrolled in other cognitive intervention programs. The recruitment period started on March 1 and concluded on October 1, 2023. The trial ended in December 2023, when the last participant completed their participation. There were not important changes to methods after trial commencement.

*Randomization and allocation*: the random allocation sequence was generated using a computer-based list of sequential numbers. The study used an open-label 2:1 randomization strategy.

A 2:1 randomization ratio was choice to support recruitment and enhance adherence among healthy volunteers by increasing the likelihood of being assigned to the intervention group. In studies involving cognitive enhancement in healthy populations, such an approach can be particularly effective in motivating participation. Given the nature of this pilot study, a larger sample size in the intervention arm allowed for a more informative preliminary assessment of feasibility and potential effects. This decision was made after careful consideration of ethical principles, study objectives, and available resources, ensuring a scientifically valid comparison between group.

Participants were assigned to either the intervention or control group in a 2:1 ratio, with blocks of three. Each block consisted of two participants allocated to the intervention group and one to the control group, ensuring that the allocation ratio was maintained. The allocation sequence was concealed by an independent researcher who was blinded to the characteristics of the participants. Volunteers were contacted in a sequential manner, and as each volunteer agreed to participate, the intervention assignment was revealed.

Those who fulfilled study requirements was invited to select a site for the in-person training sessions closest to their residence (2 sites located within the Autonomous City of Buenos Aires and 1 site in the northern metropolitan area). The data for each participant were anonymized and collected by a researcher authorized by the Research and Ethics Committee as coded data entry for subsequent analysis.

## Ethics statement and informed consent

The study was approved by the Institutional Review Board and the Local Ethics Committee. All participants provided written informed consent to be included. The study was registered in accordance with local regulations at the National Council for Scientific and Technological Research (CONICET) – Institute of Biomedical Research (BIOMED), as well as a surrogate nationally Ethics Committee. an at clinicaltrials.gov: NCT06549517 The study was registered on ClinicalTrials.gov (NCT-06549517) upon completion as the trial did not involve a clinical population. The authors confirm that all ongoing and related trials for this intervention are now registered.

## Participants

A total of 76 participants were enrolled in the study, with 56 in the intervention group and 20 in the control group. One participant from the active group was discontinued because her attention deficit hyperactivity disorder (ADHD) had not been reported at the beginning. No participants were lost to follow-up, and therefore, no intention-to-treat analysis was necessary. The analysis was conducted according to the original assigned groups. The mean age of participants was 59 years in both groups, with a slight variation in standard deviation. Most participants in both groups were male (65.0% in the control and 63.6% in the intervention group). Sixty percent of the participants in the control group and 58.2% in the intervention group were aged ≤65 years. The intervention group had a slightly greater mean years of education (18.0 years) compared to the control group (16.4 years). The baseline Addenbrooke's Cognitive Examination scores were comparable between the control (91.3) and the intervention group (91.3), indicating similar cognitive function at baseline. A small fraction of participants in the intervention group, less than 15% (n = 8), had a baseline ACE-R score of less than

85, although they did not report any cognitive complaints or show clinical manifestations of cognitive impairment in their daily life or autonomy. There were no substantial differences in clinical characteristics between the control and intervention groups. This cohort represents the general population, with typical prevalence of chronic non-communicable diseases (NCDs) such as diabetes, hypertension, and dyslipidemia **Table 1**.

## Intervention description

Mental Training Tech 24.5 (MTT24.5) is a cognitive enhancement program designed to strengthen cognitive reserve through a structured combination of novel knowledge units and multisensory instructional techniques. The program integrates two core components: (1) DATA, which represent discrete knowledge items, and (2) TECHS, which are techniques used to reinforce learning and support cognitive integration.

**DATA (knowledge units):** each DATA unit corresponds to a concise statement, no more than 15 words, representing a fact or concept derived from one of four major scientific domains: cultural, social, formal (including logic and mathematics), and biological sciences. These units are designed to be self-contained and not dependent on one another, allowing for independent cognitive engagement with each piece of content. The aim is to expose participants to knowledge that often lies outside their professional or personal expertise, thereby stimulating underused cognitive pathways. To support engagement and memory retention, each DATA unit is introduced with supplementary multimedia materials including short videos, illustrative images, contextual narratives, and emotionally engaging elements. A total of 40 DATA units are presented across the full course of the program,

**Table 1. Baseline characteristics by group.**

| Socio-demographic characteristics | Control Group (N = 20) | Intervention Group (N = 55) |
|---|---|---|
| | n/N (%) | n/N (%) |
| Sex | | |
| Male | 13/20 (65.0%) | 35/55 (63.6%) |
| Female | 7/20 (35.0%) | 20/55 (36.4%) |
| Age* | 58.9 (15.9) | 58.9 (13.3) |
| Age categorized | | |
| <=65 years | 12/20 (60.0%) | 32/55 (58.2%) |
| >65 years | 8/20 (40.0%) | 23/55 (41.8%) |
| Years of education* | 16.4 (5.2) | 18.0 (3.7) |
| Baseline ACE* | 91.3 (4.9) | 91.3 (6.2) |
| Baseline ACE categorized | | |
| <=85 | 2/20 (10.0%) | 8/55 (14.5%) |
| >85 | 18/20 (90.0%) | 47/55 (85.5%) |
| Main Clinical Characteristics | | |
| Cognitive reserve | | |
| Low | 9/20 (45.0%) | 24/55 (43.6%) |
| High | 11/20 (55.0%) | 31/55 (56.4%) |
| Diabetes | 3/20 (15.0%) | 8/55 (14.5%) |
| Hypertension | 8/20 (40.0%) | 24/55 (43.6%) |
| Dyslipidemia | 10/20 (50.0%) | 31/55 (56.4%) |
| Dementia family history | 5/20 (25.0%) | 15/55 (27.3%) |
| Use of aspirin | 5/20 (25.0%) | 8/55 (14.5%) |
| Use of statins | 6/20 (30.0%) | 20/55 (36.4%) |

*Mean and (SD) was reported.

*TECH (learning techniques):* each TECH refer to structured exercises designed to reinforce the learning of each *DATA unit* through tactile, visual, motor, and multisensory modalities. A pool of 100 TECHS was selected based on peer-reviewed evidence of their potential to stimulate neuroplasticity and enhance cognitive functioning in healthy adults from a literature review of 25 cognitive training studies published between 2000 and 2022 through search engines using terms such as *"cognitive stimulation exercises," "neuroplasticity," "adult learning,"* and *"cognitive function"* across MED-LINE (via PubMed), EMBASE, Cochrane Library, Web of Science, SCOPUS, LILACS, and Google Scholar.

Among the most frequently employed TECHS are:

- Braille Literacy (used in ~60% of pairings): Participants learn to read and write in Braille, stimulating somatosensory and spatial processing areas of the brain [26,27].

- Non-Dominant Handwriting: Tasks involving writing the DATA statement using the non-dominant hand to activate motor and executive control regions [28].

- Mirrored Bilateral Writing: Simultaneous writing with both hands—left to right with the left hand and right to left with the right—designed to engage bilateral motor coordination and interhemispheric communication.

- Multisensory Simulation: The TECHS vary in complexity and modality, some are unimodal, while others involve up to five concurrent modalities based on the evidence that multimodal stimulation enhances encoding and long-term retention. Participants could be asked to perform concurrent motor tasks (e.g., pressing a spring with one foot, moving an arm rhythmically) while decoding Braille written by another participant with closed eyes. This is intended to reinforce memory through sensorimotor integration [30].

An example of a data-tech pair is shown below:

| DATA | TECHS |
|---|---|
| The oldest man to reach the summit of Mount Everest was the Japanese climber Yuichiro Miura, who achieved this at age 80. | (1). Write the phrase "MIURA EVEREST 80" using the non-dominant hand. |
| | (2). Simultaneously write "MIURA EVEREST 80" in mirrored form with both hands (left-to-right with the left hand, right-to-left with the right hand). |
| | (3). Write "MIURA EVEREST 80" using a Braille code tablet and stylus. |
| | (4). Write in Braille a word (4–8 letters) that represents an emotion or historical context related to the DATA (e.g., courage, snow, oldest, achievement), and pass the Braille sheet to another participant, who decodes it by touch with eyes closed. |
| | (5). During the trainer's reading of statements about Miura, participants must continuously move both feet up and down while listening. When a statement is correct, they raise their non-dominant hand; if incorrect, they press a lighted button using their dominant hand. |

The association between each of the 40 DATA and its corresponding TECHS is predetermined and documented in the program's patent to ensure reproducibility and minimize variability across participant groups.

**Program delivery.** The MTT24.5 program is conducted in a presential classroom format over 12 weekly sessions totaling 24.5 instructional hours, with one certified trainer assigned to every 20 participants and additional assistants added as class size increases. Participants are required to attend at least 80% of the sessions to complete the program. The name "24.5" reflects this cumulative instructional time and draws from structures used in cognitive training studies such as the ACTIVE trial (Advanced Cognitive Training for Independent and Vital Elderly) [31]. Make-up sessions are available in case of absences to maintain program integrity and continuity of cognitive engagement.

In each class, an average of 3.5 DATA concepts are introduced, each paired with 3–4 TECHS following a standardized algorithmic format [DATA + TECHS] defined in the program's patent to ensure consistent delivery across cohorts (Fig 2).

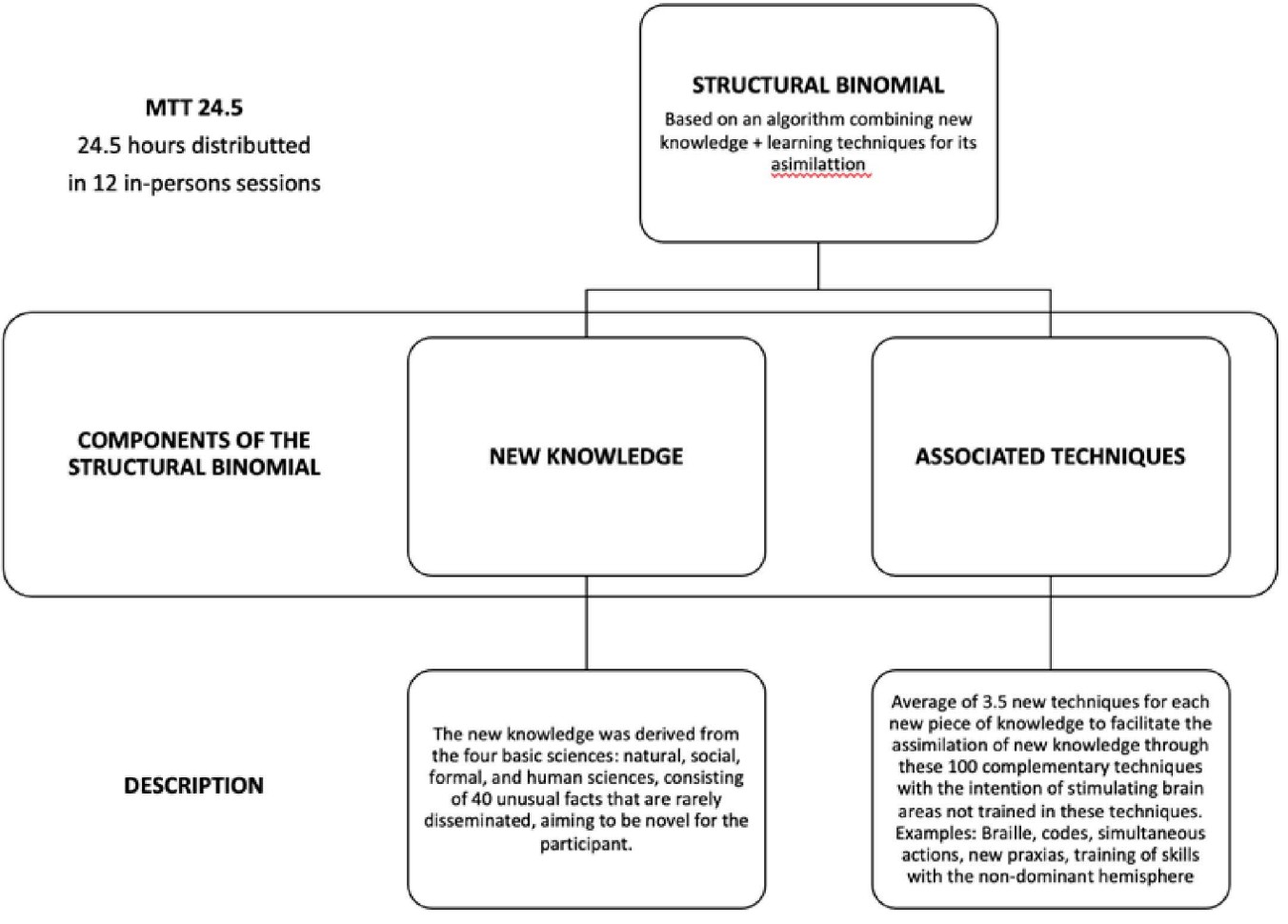

**Fig 2. MTT's structural unit defined as the combination of a new knowledge and cognitive skills for its effective learning and memorization.**
The goal is the direct stimulation of neuroplasticity and cognitive reserve through new, systematized, and structured learning.

## Evaluations

a) Addenbrooke's Cognitive Examination, ACE-R) [32] score: the ACE-R includes Mini-Mental State Examination (MMSE), and it contains a subset of tasks assessing general cognitive functioning, such as orientation, registration, attention, and recall. However, the ACE-R is more detailed and includes additional subtests for language, fluency, and visuospatial abilities, which are not fully covered by the MMS. The full revised version we used includes the MMSE and evaluation of different domains as memory, attention, language, and visuospatial skills abilities. Improvement in each cognitive ability outcome was assessed using the respective sub-dimensions of the ACE score, and the difference between post-intervention and pre-intervention sub-scores was computed. The ACE-R score ranges from 0 to 100.

b) Cognitive reserve assessment-CRS: is a test that measures participation in cognitive stimulating activities throughout a person's lifetime. Items included in the CRS assessment are scholarship, reading, playing instruments, collecting things, practicing other languages, among others [33].

## Outcomes

The primary outcome of the study was the change in global cognitive ability.

Secondary outcomes were the change in Cognitive Specific Domains: Cognitive outcomes were assessed using the sub-dimensions of the ACE-III, including memory, attention, verbal fluency, language, and visuospatial skills. The change in each domain was calculated as the difference between post-intervention and pre-intervention sub-scores. The sub-score ranges are as follows: memory (0–26), attention (0–8), verbal fluency (0–14), language (0–16), and visuospatial skills (0–12). There were no changes to trial outcomes after the trial commenced.

## Statistical analyses

*Statistical plan:* The study protocol, statistical analysis plan according CONSORT guidelines are available in the complementary file SAP.

Descriptive Statistics: Socio-demographic and clinical characteristics of participants were summarized using means and standard deviations (SD) for continuous variables, and frequencies and percentages for categorical variables.

A sample size of 75 subjects will provide a power of at least 80% to detect a minimum difference of 4 points in the score improvement between the two groups, assuming a maximum standard deviation of 5.4 points. This effect size of 0.80 used in our sample size calculation was based on prior literature, particularly the methodology and effect size interpretation used in the ACTIVE trial (Advanced Cognitive Training for Independent and Vital Elderly). Although the observed effects in ACTIVE ranged from small to moderate (e.g., 0.23 for reasoning and 0.66 for speed), we considered an effect size of 0.80 as a conservative target to detect a meaningful difference with sufficient power, assuming a best-case scenario for the intervention's impact. Additionally, we considered the characteristics of our primary cognitive outcome measure. For example, in the Addenbrooke's Cognitive Examination (ACE), a 1-point difference is clinically relevant for identifying cognitive decline. Therefore, assuming an intervention could achieve at least a moderate to large improvement (corresponding to 4 points, SD = 5.4, d = 0.8) was both clinically reasonable and consistent with previous literature on cognitive training effects.

**Impact Assessment:** The effect size was calculated as the difference in mean improvement between the intervention and control groups. Cohen's d statistic was computed to determine standardized effect sizes, facilitating comparisons across different measures. Cohen's d is defined as the mean difference between groups divided by the pooled standard deviation [34]. Cohen's statistic ((Mean 1 – Mean 2)/ Pooled SD) can be calculated both when the groups have the same size and when the groups have different sizes (Cohen, 1988. *Statistical Power Analysis for the Behavioral Sciences*). We also applied linear models were applied to each cognitive domain (global, memory, attention, orientation, verbal fluency, language, and visuospatial skills). Each model used the improvement score as the dependent variable and group (intervention vs. control) as the independent variable.

**Covariate Analysis:** To explore potential response phenotypes, linear models were used for each covariate of interest (e.g., age, gender, medical history, medication use, family history of dementia, baseline cognitive reserve score, and use of aspirin and statins). The primary outcome was modeled with the covariate, group, and their interaction term to assess whether the intervention effect varied across different covariate groups.

**Statistical Testing:** All tests were two-sided with a significance level of 5%. Confidence intervals were 95% and two-sided. P-values were reported to three decimal places if ≥0.001; values <0.001 were reported as "<0.001". Given the exploratory nature of this pilot study, no corrections for multiple comparisons were applied. This approach was chosen to prioritize sensitivity to potential signals that could inform future confirmatory research. Mean and standard deviation were reported to one decimal place more than the original data. Quantiles (median, minimum, maximum) were reported with the same precision as the original data. Estimated parameters not on the same scale as raw observations were reported to significant figures.

**Software:** Statistical analyses were performed using R version 4.3.0 [35].

## Results

### Changes in cognitive performance

The average baseline score and (standard deviation) for the global cognitive ability in the control group and in the intervention were 91.3 (±4.9) and 91.3 (±6.2) respectively. The intervention group had a significant improvement from baseline in the global cognitive ability compared to the control group, with an average of 4.6 points from baseline compared to a decrease of 0.5 points in the control group. The non-standardized effect size was 5.14 (95% CI: 3.16 to 7.11) and the standardized effect size was 1.33 (95% CI: 0.77 to 1.90), indicating a statistically significant positive impact of the intervention on global cognitive function When comparing the standardized effect sizes across different domains, memory exhibited the highest effect size, equal to 0.72 (95% CI: 0.19 to 1.25) (**Table 2** and **Fig 3A** and **3B***).

**Table 2. Effect of MTT 24.5 program on cognitive abilities outcomes.**

| Outcome | Control Group (N = 20) | Intervention Group (N = 55) | P-value |
|---|---|---|---|
| PRIMARY OUTCOME | | | |
| **Global cognitive ability** | | | **<0.001** |
| Score at baseline, mean (±SD) | 91.3 (4.9) | 91.3 (6.2) | |
| Improvement from baseline, mean | −0.5 | 4.6 | |
| Non-standardized effect size (95% CI) * | – | 5.14 (3.16; 7.11) | |
| Standardized effect size (95% CI) ** | – | 1.33 (0.77; 1.90) | |
| SECONDARY OUTCOMES | | | |
| **Attention cognitive ability** | | | **0.055** |
| Score at baseline, mean (±SD) | 8.0 (0.0) | 7.7 (0.7) | |
| Improvement from baseline, mean | 0.0 | 0.3 | |
| Non-standardized effect size (95% CI) * | – | 0.27 (−0.001; 0.55) | |
| Standardized effect size (95% CI) ** | – | 0.51 (−0.02; 1.04) | |
| **Memory cognitive ability** | | | **0.007** |
| Score at baseline, mean (±SD) | 21.2 (3.5) | 21.1 (3.9) | |
| Improvement from baseline, mean | 0.3 | 2.4 | |
| Non-standardized effect size (95% CI) * | – | 2.12 (0.61; 3.63) | |
| Standardized effect size (95% CI) ** | – | 0.72 (0.19; 1.25) | |
| **Verbal fluency cognitive ability** | | | **0.018** |
| Score at baseline, mean (±SD) | 12.9 (1.3) | 12.7 (2.0) | |
| Improvement from baseline, mean | −0.3 | 0.7 | |
| Non-standardized effect size (95% CI)* | – | 1.00 (0.19; 1.81) | |
| Standardized effect size (95% CI) ** | – | 0.63 (0.10; 1.17) | |
| **Visuospatial skill's cognitive ability** | | | **0.024** |
| Score at baseline, mean (±SD) | 14.2 (1.7) | 15.1 (1.2) | |
| Improvement from baseline, mean | −0.2 | 0.5 | |
| Non-standardized effect size (95% CI) * | – | 0.73 (0.11; 1.35) | |
| Standardized effect size (95% CI) ** | – | 0.60 (0.07; 1.13) | |

Abbreviations: CI: confidence interval; SD: standard deviation.

*Non-standardized effect size defined as the difference between the mean improvement on the intervention group and the mean improvement on the control group **The standardized effect size defined as the difference between the mean improvement on the intervention group and the mean improvement on the control group, divided by the pooled standard error (Cohen's d Statistic). Noted that *P-values are unadjusted for multiple comparisons due to the exploratory nature of the study.*

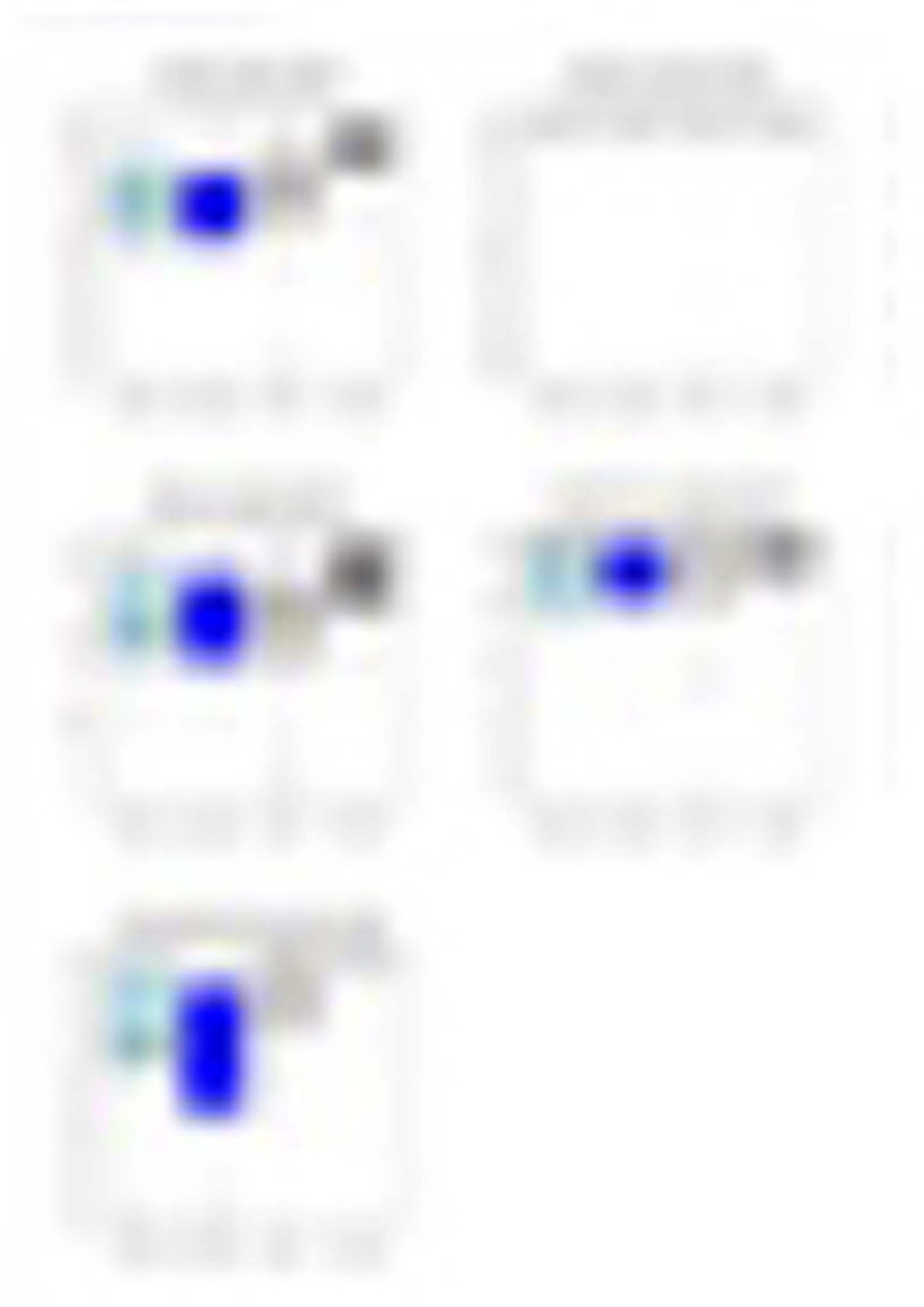

**Global cognitive ability**

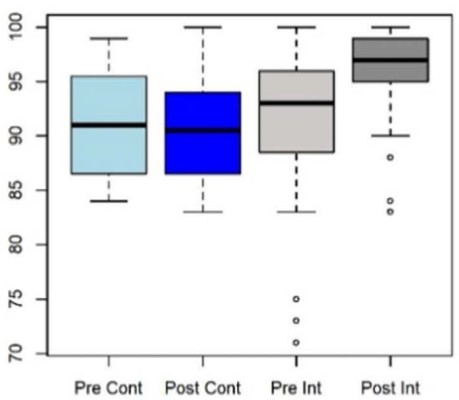

**Attention cognitive ability**

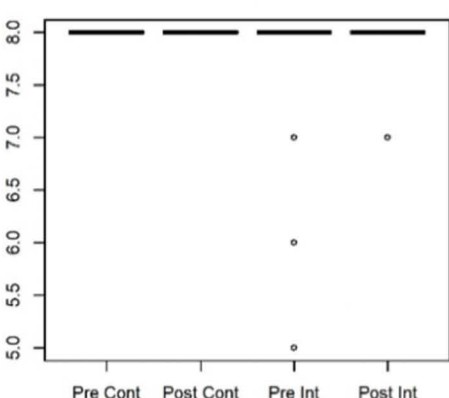

**Memory cognitive ability**

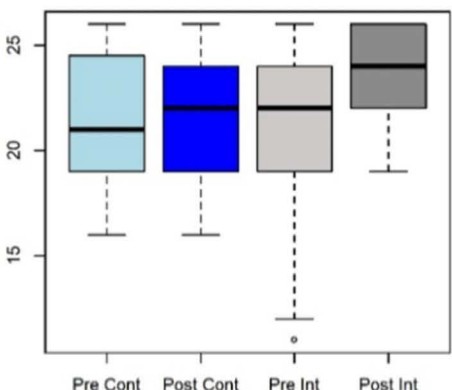

**Verbal fluency cognitive ability**

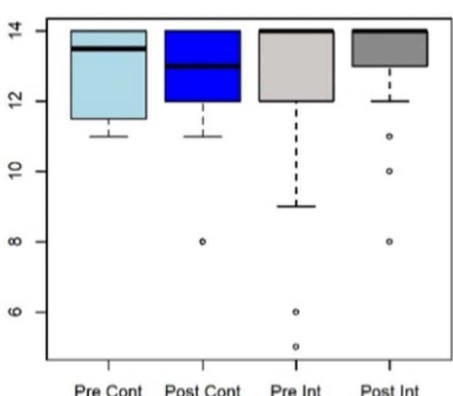

**Visuospatial skills cognitive ability**

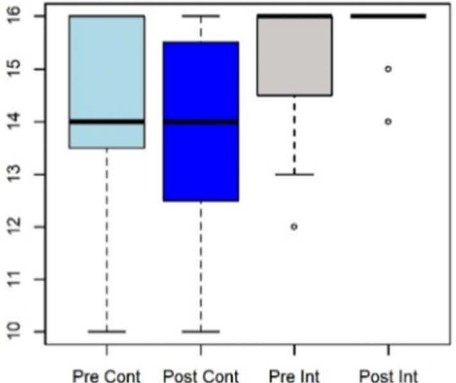

**Fig 3. A. Mean change from baseline on Cognitive Ability Scores.** The lines represent the 95% confidence interval around each mean, constructed based on the standard error of the mean (SE), the bars depict the mean change score for each group. B. Distribution of the cognitive abilities at pre-and post-intervention.

The impact of the MTT 24.5 program on global cognitive ability across clinical covariates defined by their baseline characteristics and clinical factors was assessed. There was a consistent impact of the intervention across these various subgroups or phenotypes considered. Participants with a baseline ACE score less than 85 showed significantly greater improvement compared to those with a baseline ACE score >85 (p-value: 0.003). (**Fig 4**).

## Discussion

The results of this study demonstrate the effectiveness of the Mental Training Tech 24.5 (MTT24.5) cognitive stimulation program in enhancing cognitive function among healthy adults from the general population. The program significantly improved global cognitive performance and enhanced five of the six cognitive domains evaluated, suggesting its potential as an effective tool for cognitive enhancement.

According to Christopher Hertzog's theory of cognitive development, cognitive performance has an upper limit, or Cognitive Development Potential (DCP), which is influenced by various factors such as cognitive enrichment, genetics, and health [36].The significant improvements observed in participants following the MTT24.5 program could be a result of its structured approach to learning, which enhances cognitive potential within this framework. By promoting the acquisition of novel, unrelated knowledge, the program stimulates neuroplasticity, improving brain function across different cognitive domains. The role of neuroplasticity in cognitive enhancement is well-established as it is shaped by a combination of biological, environmental, and behavioral factors. MTT24.5 may help optimize cognitive development and mitigate the effects of age-related cognitive decline.

In the context of cognitive function enhancement, the application of Network Control Theory (NCT) [37] offers a useful framework for understanding how stimulating new learning experiences can lead to improved cognitive outcomes. This idea is further elaborated by Bassett and Sporns [38], who posit that brain networks function as complex systems that are governed by controllable states, where certain brain regions serve as hubs that can influence the overall function of the network. This concept aligns with the idea that targeted stimuli, such as new knowledge acquisition and the associated learning techniques, can modulate and strengthen the functional connectivity of these hubs, ultimately improving cognitive outcomes.

When individuals engage in novel learning experiences, such as acquiring new data or knowledge across different domains (e.g., cultural, biological, social sciences), the brain's neural pathways are challenged and reorganized. This process, driven by cognitive engagement, activates and strengthens brain networks that are essential for functions like memory, attention, and processing speed. The brain's ability to adapt and reorganize in response to these stimuli is a manifestation of neuroplasticity, which is known to be influenced by learning tasks that involve both novelty and complexity.

For example, incorporating techniques such as Braille literacy, an activity that involves tactile learning and motor coordination, provides a multi-dimensional challenge that engages various brain areas. Braille literacy, as a form of cognitive training, has been shown to enhance sensory integration and motor control, fostering neural connections across multiple brain regions. Through this engagement, the internalization of new knowledge, combined with the specific technique of Braille reading and writing, could facilitate the improvement of cognitive functions like attention, memory, and processing speed.

As outlined in previous studies, such as the Nun Study [16] and the London Taxi Drivers Study [19], lifelong learning and continuous cognitive engagement have been shown to protect cognitive function in aging individuals. These studies provide strong evidence that sustained intellectual stimulation—such as the acquisition of new knowledge and skills—plays a crucial role in building cognitive reserve. By enhancing the brain's adaptability and functional connectivity, lifelong learning helps preserve cognitive abilities like memory, attention, and processing speed, even in the face of aging.

The MTT24.5 methodology, which is based on explicit learning tasks, stands in contrast to research by J.R. Anderson et al., who demonstrated that identical skills could emerge from both explicit instruction and discovery-based learning [39]. Our findings suggest that future evaluations of MTT24.5 should explore whether the program's effectiveness is influenced

| Subgroup | Mean Bas | Score | Mean | Improvement | | MD (95% CI) | P-value |
|---|---|---|---|---|---|---|---|
| | Control | Intervention | Control | Intervention | | | |
| Sex | | | | | | | 0.312 |
| Female (n=48) | 91.0 | 92.2 | -0.54 | 3.83 | | 4.37 [1.95, 6.78] | |
| Male (n=27) | 91.9 | 89.8 | -0.43 | 6.05 | | 6.48 [3.21, 9.75] | |
| Age | | | | | | | 0.826 |
| <65 years (n=44) | 91.1 | 90.8 | 0.00 | 4.97 | | 4.97 [2.39, 7.54] | |
| >=65 years (n=31) | 91.6 | 92.0 | -1.25 | 4.17 | | 5.42 [2.30, 8.55] | |
| Years of Education | | | | | | | 0.081 |
| <=12 years (n=6) | 89.4 | 73.0 | -1.40 | 11.00 | | 12.40 [4.19, 20.61] | |
| >12 years (n=69) | 91.9 | 91.7 | -0.20 | 4.52 | | 4.72 [2.53, 6.91] | |
| Cognitive Reserve | | | | | | | 0.697 |
| Low (n=34) | 89.9 | 89.0 | -0.67 | 4.92 | | 5.58 [2.59, 8.57] | |
| High (n=42) | 92.5 | 93.1 | -0.36 | 4.42 | | 4.78 [2.10, 7.47] | |
| Baseline ACE | | | | | | | 0.003 |
| <=85 (n=11) | 84.0 | 79.9 | 0.00 | 11.62 | | 11.63 [7.02, 16.23] | |
| >85 (n=65) | 92.1 | 93.3 | -0.56 | 3.45 | | 4.00 [2.39, 5.62] | |
| Diabetes history | | | | | | | 0.955 |
| No (n=64) | 91.5 | 91.8 | -0.59 | 4.57 | | 5.16 [3.00, 7.33] | |
| Yes (n=11) | 90.0 | 88.5 | 0.00 | 5.00 | | 5.00 [-0.18, 10.18] | |
| Hypertension history | | | | | | | 0.439 |
| No (n=43) | 92.2 | 91.7 | -0.75 | 5.06 | | 5.81 [3.23, 8.40] | |
| Yes (n=32) | 90.0 | 90.8 | -0.12 | 4.08 | | 4.21 [1.10, 7.32] | |
| Dyslipidemia history | | | | | | | 0.972 |
| No (n=34) | 93.0 | 91.6 | -0.40 | 4.71 | | 5.11 [2.22, 7.99] | |
| Yes (n=41) | 89.6 | 91.1 | -0.60 | 4.58 | | 5.18 [2.39, 7.97] | |
| Myocardial infarction | | | | | | | 0.470 |
| No (n=69) | 91.6 | 91.5 | -0.50 | 4.45 | | 4.95 [2.87, 7.03] | |
| Yes (n=6) | 88.5 | 89.5 | -0.50 | 7.00 | | 7.50 [0.94, 14.06] | |
| Previous cerebrovascular accident | | | | | | | 0.853 |
| No (n=66) | 91.6 | 92.0 | -0.44 | 4.60 | | 5.05 [2.92, 7.18] | |
| Yes (n=8) | 86.0 | 87.0 | -1.00 | 4.86 | | 5.86 [-2.39, 14.11] | |
| Sleep apnea syndrome | | | | | | | 0.285 |
| No (n=70) | 91.7 | 91.8 | -0.47 | 4.33 | | 4.81 [2.81, 6.80] | |
| Yes (n=5) | 84.0 | 85.2 | -1.00 | 8.50 | | 9.50 [1.19, 17.81] | |
| Dementia family history | | | | | | | 0.403 |
| No (n=55) | 91.9 | 91.9 | -0.53 | 4.08 | | 4.61 [2.34, 6.88] | |
| Yes (n=20) | 89.4 | 89.8 | -0.40 | 6.13 | | 6.53 [2.66, 10.41] | |
| Use of aspirin | | | | | | | 0.175 |
| No (n=62) | 91.1 | 92.2 | -0.67 | 4.06 | | 4.73 [2.57, 6.89] | |
| Yes (n=13) | 92.0 | 86.1 | 0.00 | 8.00 | | 8.00 [3.85, 12.15] | |
| Use of statins | | | | | | [, ] | 0.807 |
| No (n=49) | 91.2 | 91.6 | -0.79 | 4.49 | | 5.27 [2.85, 7.69] | |
| Yes (n=26) | 91.5 | 90.8 | 0.17 | 4.90 | | 4.73 [1.18, 8.29] | |
| Overall | 91.3 | 91.3 | -0.50 | 4.64 | | 5.14 [3.16, 7.11] | <0,001 |

-15 -10 -5 0 5 10 15 20 25
Mean Difference

← Favours Control Group Favours Intervention Group →

**Fig 4. Clinical covariates and cognitive response phenotypes.** Forest plot to show the moderating effects of clinical and demographic covariates are corrected for multiple comparisons when assessing all these moderators.

by the mode of learning—whether explicit guidance or more implicit discovery-based strategies lead to the most significant cognitive improvements. This distinction may be particularly important in the application of MTT24.5 to complex skills such as Braille literacy, where different modes of learning may have varying impacts on effectiveness.

Further, the neuroplastic benefits of cognitive training extend beyond traditional cognitive domains. Braille learning, for instance, has been shown to reorganize brain regions associated with sensory and motor functions, even in sighted individuals. Studies on Braille readers have demonstrated reorganization of the visual cortex and other sensory areas, supporting the idea that cognitive stimulation can induce significant neuroplastic changes across sensory systems [40]. This reinforces the potential of MTT24.5 as a tool for inducing broad neuroplastic changes in response to novel learning experiences.

Although most participants had baseline cognitive scores within the normal range, a small subset showed lower baseline scores. It is important to note that MTT24.5 was equally effective across the full spectrum of baseline cognitive performance, suggesting that the program may have universal applicability for cognitive enhancement.

Our results are consistent with the findings of previous studies using online cognitive training platforms such as Lumosity [16]. Both MTT24.5 and Lumosity demonstrate that cognitive training can improve core cognitive abilities, including processing speed, working memory, and fluid reasoning. However, there are critical differences between the two programs. While **Lumosity** uses games to target specific cognitive skills, MTT24.5 emphasizes the assimilation of new knowledge, with both gamified and non-gamified exercises designed to consolidate this knowledge into long-term memory. In this respect, Lumosity's focus on skill training is only one component of the broader neuroplastic stimulation aimed at by MTT24.5. Future research is needed to evaluate the relative contributions of the program's two components to its overall effectiveness. A study comparing data, technology, and a combination of both would provide valuable insights into their individual contributions.

In terms of limitations, the cognitive test battery used in this study could be expanded to mitigate potential bias, as the ACE-R test, like the MMSE and MoCA, is mainly a screening tool for cognitive impairment. However, despite these limitations, the findings indicate that the MTT24.5 program has significant effects across different baseline cognitive scores, with meaningful improvements observed in both lower and higher baseline groups. Additionally, while the study sample was recruited from the general population, the relatively small sample size and the lack of a population-based sampling method may limit the generalizability of the results. Future studies should address these limitations by employing larger, more diverse samples to assess the broader applicability of these findings.

## Conclusion

This study provides compelling evidence for the effectiveness of the MTT24.5 cognitive stimulation program in enhancing cognitive function among healthy adults from the general population. The program's approach—combining the assimilation of novel knowledge with structured learning techniques to stimulate neuroplasticity—resulted in significant improvements across multiple cognitive domains, particularly in memory. Given these promising results, we suggest that future studies explore whether cognitive training should be integrated into recommendations for promoting cognitive health, alongside physical activity, to prevent age-related cognitive decline.

Furthermore, while this study demonstrates significant cognitive improvements, additional research is needed to explore the long-term effects of the program. Specifically, studies should evaluate whether these cognitive improvements persist over time and whether MTT24.5 could delay the onset of age-related cognitive decline, including the risk of dementia. In addition, further investigations comparing different components of the program (data versus technology) will be critical to understanding how each contributes to its effectiveness.

## Supporting information

**S1 File. Protocol and Statistical Analysis Plan (SAP).** This file includes the original study protocol and the statistical plan that outlines the study objectives, methodology and prespecified statistical methods.
(PDF)

**S2 File. Original Study Protocol (Spanish).** This file contains the original version of the study protocol in Spanish, as approved by the Institutional Review Board (IRB) prior to study initiation.
(PDF)

**S3 File. Data underlying the findings.** This file contains the de-identified dataset used for all analyses presented in the manuscript. The data correspond to the variables and outcomes described in the study protocol and statistical analysis plan.
(XLSX)

**S4 File. Code References for lecture of S3 dataset.**
(PDF)

## Acknowledgments

We gratefully acknowledge the contributions of K. Ulens (MD, Barcelona) for her collaboration in the database; M. Berrueta, N. Castellana, and L. Gibbons (IECS, Argentina) for their unvaluable contribution for the statistical analysis plan; and Diego García Villanueva for his mentorship.

## Author contributions

**Conceptualization:** Carol Kotliar, Ricardo Jauregui, Oscar Cingolani.

**Formal analysis:** Lisandro Olmos.

**Investigation:** Carol Kotliar, Martín Koretzky, Ricardo Jauregui, Tomás Delía.

**Methodology:** Carol Kotliar, Ricardo Jauregui.

**Project administration:** Tomás Delía.

**Validation:** Carol Kotliar.

**Writing – original draft:** Carol Kotliar.

**Writing – review & editing:** Carol Kotliar, Lisandro Olmos, Tomás Delía, Oscar Cingolani.

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
