## [Decision Letter · Decision Letter 0]

4 Mar 2025

Dear Dr.  Kotliar,

Thank you for submitting your manuscript to PLOS ONE. After careful consideration, we feel that it has merit but does not fully meet PLOS ONE’s publication criteria as it currently stands. Therefore, we invite you to submit a revised version of the manuscript that addresses the points raised during the review process.

We look forward to receiving your revised manuscript.

Kind regards,

Roya Khanmohammadi, Ph.D

Academic Editor

PLOS ONE

Journal Requirements:

3. In the online submission form, you indicated that “All files are available upon justified request.”.

5. Please amend your manuscript to include your abstract after the title page.

Reviewers' comments:

Reviewer's Responses to Questions

**Comments to the Author**

1. Is the manuscript technically sound, and do the data support the conclusions?

Reviewer #1: Yes

Reviewer #2: Yes

2. Has the statistical analysis been performed appropriately and rigorously?

Reviewer #1: Yes

Reviewer #2: Yes

3. Have the authors made all data underlying the findings in their manuscript fully available?

Reviewer #1: Yes

Reviewer #2: No

4. Is the manuscript presented in an intelligible fashion and written in standard English?

Reviewer #1: Yes

Reviewer #2: Yes

Reviewer #1: The authors have presented a highly relevant cognitive stimulation program (MTT24.5) aimed to improve global cognitive performance in healthy, older adults. While this paper is overall well written, there are theoretical and methodological concerns, which lead me to my recommendation of “revise and resubmit”.

Introduction:

Throughout the entire introduction, the authors include statements, commentary, and previous studies without citations. Given that the reference list includes more citations than what was used in this section, perhaps some were left out.

The introduction provides vague information on previous studies that have shown evidence for the benefits of engaging in multiple-skill learning interventions to enhance neuroplasticity, and improve cognitive reserve in healthy adults. While much of the introduction states that learning interventions improve cognitive skills, there is a lack of literature that addresses the key objective in this study: testing a program where learning multiple skills can enhance multiple brain regions. Rather than broadly discussing how learning interventions improve ‘neuroplasticity’ and ‘cognitive function’, address how these interventions have improved the cognitive functions measured in this study, such as memory, attention, verbal fluency, language, and visuospatial skills. Providing this extra clarity can strengthen the introduction.

Although the importance of investigating the gap in evidence on the effectiveness of cognitive training interventions in healthy older adults is addressed, there is a lack of literature focused on the authors’ choice of subjects: healthy individuals, rather than individuals with cognitive impairments. A stronger focus on studies involving healthy older adults would better support the study’s rationale and help formulate a more specific hypothesis.

The authors state that this pilot study aims to develop a hypothesis to validate the effectiveness of the MTT24.5 for improving cognitive performance, however this approach may hinder the validity of this technique as a more precise hypothesis could have been formulated based on the previous success of cognitive learning interventions with similar approaches in healthy older adults.

Method:

While the pairs (DATA and TECHS) are introduced and methodologically explained thoroughly, it remains theoretically unclear why the authors chose these pairs and how they both contributed to the study’s objective.

A few minor methodological details are missing in this design. For example, how were these pairs administered, executed, and monitored? Were the pairs performed digitally or in a classroom setting? It is stated that this is an in-person study, however there is a lack of information regarding the study’s procedures.

Results:

The results section of the paper includes the demographic information of the participants. However, this table would be more appropriate if presented in the Method section.

Reviewer #2: Introduction

Providing a deeper explanation of how MTT24.5 promotes neuroplasticity would enhance the manuscript. Discussing theoretical frameworks, such as network control theory, could offer valuable insights into the underlying mechanisms.

The last sentence of the introduction states that "This cohort represents the general population." Is this study a clinical trial or a cohort study?

Method

Why is the ratio of the intervention group to the control group considered to be 2:1?

**Do you want your identity to be public for this peer review?** For information about this choice, including consent withdrawal, please see our Privacy Policy

Reviewer #1: No

Reviewer #2: No

---

## [Author Response · Author response to Decision Letter 1]

11 Apr 2025

Dear Reviewers, #1 and #2,

We would like to sincerely thank you for the time and effort you have dedicated to reviewing our manuscript. We deeply appreciate your thoughtful comments and feedback. We are very hopeful to communicate our findings to our colleagues through the prestigious scientific journal, PLOS ONE.

Below, we respond to each of the suggestions, comments, and recommendations from the reviewers. We also confirm that we have made the necessary changes and additions to meet all the requirements outlined in your evaluation letter.

We believe these changes address all the concerns raised and have significantly improved the manuscript. We are grateful for your guidance throughout this process and look forward to your feedback.

Thank you once again for your time and support.

Sincerely,

Carol Kotliar, PhD

Responses to Reviewer #1

1. Comment: “Throughout the entire introduction, the authors include statements, commentary, and previous studies without citations. Given that the reference list includes more citations than what was used in this section, perhaps some were left out.”

Response: Following your observation, we have included the appropriate citations for all statements, commentary, and previous studies as requested. The reference list has been reviewed and updated to complement this valuable observation. (#See pages 3 to 5 for the updated introduction with the corresponding citations from references 1 to 28, which can be found in the references section on pages 24 to 27)

2. Comment: “Rather than broadly discussing how learning interventions improve ‘neuroplasticity’ and ‘cognitive function,’ address how these interventions have improved the cognitive functions measured in this study, such as memory, attention, verbal fluency, language, and visuospatial skills. Providing this extra clarity can strengthen the introduction.”

Response: We consider your comment extremely valuable in strengthening the focus of the introduction on our study's specific scope. We have restructured and enhanced the introduction, providing a foundation with the corresponding citations that address how learning interventions have improved the specific functions measured in this study. (#See the modifications on pages 3 to 5 throughout the entire revised introduction)

3. Comment: “A stronger focus on studies involving healthy older adults would better support the study’s rationale and help formulate a more specific hypothesis. The authors state that this pilot study aims to develop a hypothesis to validate the effectiveness of the MTT24.5 for improving cognitive performance, however, this approach may hinder the validity of this technique as a more precise hypothesis could have been formulated based on the previous success of cognitive learning interventions with similar approaches in healthy older adults.”

Response: We also appreciate this valuable comment. We have now focused more on studies involving healthy adults in the introduction and emphasized this condition among of the adults included in the study in the formulation of the hypothesis. Additionally, we thank the reviewer for highlighting this specific characteristic of our population, which could undoubtedly be considered for future prevention studies to be applied in the general population. Given the increased life expectancy, brain stimulation or training could be a potential recommendation to contribute to the reduction of future cognitive decline risk. (See the modifications on pages 3 to 5 throughout the entire revised introduction)

4. Comment: “While the pairs (DATA and TECHS) are introduced and methodologically explained thoroughly, it remains theoretically unclear why the authors chose these pairs and how they both contributed to the study’s objective.”

Response: Thank you for your valuable question. The origin of the MTT design, utilizing binomial units of "DATA" and "TECHS," was proposed by the lead author of the manuscript, Carol Kotliar, PhD, who also registered it as an intellectual property. The core idea was to create a structured, progressive learning system that would introduce new knowledge over time. The term "DATA" was chosen to describe these new units of knowledge, each represented as a simple statement encapsulated in a concise sentence. The original concept was to design the knowledge in such a way that each "DATA" unit could stand independently without requiring interconnection with others. To achieve this, the knowledge was carefully selected from four main scientific fields: cultural, formal (including logic and mathematics), biological, and social sciences. This diversified approach aimed to stimulate various areas of the brain, with some of the knowledge deliberately positioned outside the participant's usual domain of expertise. Given that it is unlikely an individual would have a professional or personal interest in all these areas, this strategy ensured broader cognitive engagement, pushing participants beyond their comfort zones and established knowledge base.

Data for each unit was carefully sourced using search engines focusing on these four fields. Each "DATA" point was distilled into a single, easily memorable sentence. To enhance the engagement and accessibility of the material, additional resources such as videos, images, anecdotes, and emotional cues were incorporated alongside the core content.

In parallel, the term "TECHS" was coined to describe the methodologies used to facilitate the learning of the "DATA." These techniques were selected based on their documented association with neuroplastic changes in the brain, as evidenced by structural and functional studies in the scientific literature. Among the various techniques explored, Braille literacy was identified as particularly effective, with approximately 60% of the "TECHS" incorporating Braille reading and writing exercises. This inclusion was grounded in the evidence of its neuroplastic benefits, stimulating brain regions associated with sensory integration, motor control, and memory retention.

Participants in the MTT program developed Braille reading and writing skills as part of the curriculum, as well as the ability to perform mirrored writing with both hands simultaneously. The content written and read always corresponded to the new "DATA," reinforcing the memorization process throughout the activities.

The dual structure of the MTT program comprising "DATA" (the new knowledge) and "TECHS" (the methods for learning) was designed to promote dynamic cognitive engagement. By offering a varied and structured approach, it aimed to enhance neuroplasticity through the stimulation of multiple brain regions and encourage the development of novel cognitive pathways in healthy adults. (A summarized expansion of this justification for the Data-Tech pair has been included in the Methods section on pages 11 and 12)

5. Comment: “A few minor methodological details are missing in this design. For example, how were these pairs administered, executed, and monitored? Were the pairs performed digitally or in a classroom setting?”

Response: The binomial Data-Tech pair is presented in a classroom setting. The sessions are taught in person by a certified trainer who uses standardized materials, including videos, images, oral descriptions, and various resources such as sheets of paper, pencils, Braille tablets with punchers, a luminous button, a semi-baked ball for foot pressure, a calculator, Wisconsin cards, the Stroop Color Test, and other tools integrated into the techniques. Each participant is assigned specific materials. There is one trainer for every 20 participants, with an assistant; the number of assistants increases as the class size grows. To complete the full MTT, participants must attend 12 consecutive sessions.

The program is executed sequentially, with all data sets organized for each class. Each session combines one piece of knowledge from each of the four scientific fields cultural, social, formal, and biological sciences; administered in a specific order across the 12 sessions, along with their corresponding techniques. The trainer explains each piece of data to all participants during each session, followed by the techniques associated with that data. After completing the first data-tech pair, the process progresses to the second, then the third, continuing in this structured manner throughout each session.

A participant is considered to have completed the program if they attend at least 80% of the sessions. If a participant is unable to attend a session, they are encouraged to make it up in a later class, but they cannot miss any classes without attending a make-up session. (#Following your recommendation, the description is expanded in the Methods section on page 10.)"

III. Responses to reviewer #2

1. Comment: “Providing a deeper explanation of how MTT24.5 promotes neuroplasticity would enhance the manuscript. Discussing theoretical frameworks, such as network control theory, could offer valuable insights into the underlying mechanisms.”

Response: To address the reviewer’s request, we provided a deeper explanation of how the Mental Training Tech 24.5 (MTT24.5) program may promote neuroplasticity by integrating the theoretical framework of Network Control Theory (NCT). This analysis will link the effects of the program on cognitive functions (attention, memory, verbal fluency, visuospatial skills, and processing speed) to the principles of NCT. We appreciate this valuable suggestion, which has been incorporated into the manuscript. Considering that this perspective adds depth to the interpretation of the results, we believe that its inclusion in the discussion strongly enhances the impact of these concepts.

We believe this significantly enriches our manuscript. (#See the pages 17 and 18)

2. Comment: “The last sentence of the introduction states that 'This cohort represents the general population.' Is this study a clinical trial or a cohort study?”

Response: This was certainly necessary to modify. We have updated the last sentence of the introduction, clarifying that the study is a clinical trial, not a cohort study. (#See the correction in introduction section at page 5)

Section: Method

3. Comment: “Why is the ratio of the intervention group to the control group considered to be 2:1?”

Response: A 2:1 randomization ratio was chosen to facilitate recruitment among healthy volunteers, offering a higher chance of receiving the Intervention. In the case of potentially improving cognitive function, this ratio, which allocates more 'healthy' participants to the experimental treatment in a short period, can promote better adherence and recruitment. This approach was particularly relevant given that this was a pilot study. By using this design, a larger sample size in the intervention group was achieved. The decision was made after careful consideration of study resources and ethical guidelines to ensure a fair and scientifically sound comparison between groups.

Section: Results

4. Comment (d) of Reviewer #2: “The results section of the paper includes the demographic information of the participants. However, this table would be more appropriate if presented in the Methods section.”

Response: The table describing the demographic information of the participants has been moved to the Methods section as suggested. (#see page 9)

---

## [Decision Letter · Decision Letter 1]

22 Apr 2025

Dear Dr. Kotliar,

Thank you for submitting your manuscript to PLOS ONE. After careful consideration, we feel that it has merit but does not fully meet PLOS ONE’s publication criteria as it currently stands. Therefore, we invite you to submit a revised version of the manuscript that addresses the points raised during the review process.

While prior concerns have been largely addressed, a few minor revisions are still needed:

**Introduction** : Please reference platforms like *Lumosity* earlier, as this supports the novelty of your work.**Methods** : Clarify and streamline the explanation of the DATA and TECHS components to improve readability and reduce redundancy.

We look forward to receiving your revised manuscript.

We look forward to receiving your revised manuscript.

Kind regards,

Roya Khanmohammadi, Ph.D

Academic Editor

PLOS ONE

Journal Requirements:

Reviewers' comments:

Reviewer's Responses to Questions

**Comments to the Author**

Reviewer #1: All comments have been addressed

Reviewer #2: All comments have been addressed

2. Is the manuscript technically sound, and do the data support the conclusions?

Reviewer #1: Yes

Reviewer #2: Yes

3. Has the statistical analysis been performed appropriately and rigorously?

Reviewer #1: Yes

Reviewer #2: Yes

4. Have the authors made all data underlying the findings in their manuscript fully available?

Reviewer #1: Yes

Reviewer #2: Yes

5. Is the manuscript presented in an intelligible fashion and written in standard English?

Reviewer #1: Yes

Reviewer #2: Yes

Reviewer #1: The author’s have presented a highly relevant cognitive stimulation program (MTT24.5), displaying improved cognitive abilities in healthy adults. Many of the previous concerns of the manuscript were resolved, however there are a few minor theoretical concerns, which lead me to my recommendation of “revise and resubmit”.

Introduction:

At the end of the discussion, the authors mention that their findings are consistent with online cognitive training platforms such as Lumosity. However, this is the first time the authors mention this online cognitive training platform, only at the end of the paper. Considering both Luminosity and the current study highlight similar benefits of enhanced cognitive abilities, it would strengthen the introduction and display novelty to the current study if Lumosity were introduced before the current study.

Methods:

The authors provided information on how the DATA and TECHS components are used for the intervention; however, the overall organization of these concepts throughout the methods section is not optimal. Especially when initially introducing the components, they are abstract and contain numerous redundancies. As these two components are central to the intervention, it is crucial that they are explained more clearly.

Reviewer #2: I appreciate the authors’ efforts — the comments have been thoughtfully addressed and implemented effectively.

**Do you want your identity to be public for this peer review?** For information about this choice, including consent withdrawal, please see our Privacy Policy

Reviewer #1: No

Reviewer #2: No

---

## [Author Response · Author response to Decision Letter 2]

20 May 2025

The minor changes requested by Reviewer #1 were implemented, including the mention of the study on the cognitive platform Lumosity in the Introduction, and a rewritten description of the intervention in the Methods section to enhance clarity. Due to the inclusion of the Hardy et al. study in the Introduction, its position in the reference list was updated, which required adjusting the overall numbering of references accordingly.

---

## [Editor Report · Decision Letter 2]

28 May 2025

Dear Dr. Kotliar,

Thank you for submitting your manuscript to PLOS ONE. After careful consideration, we feel that it has merit but does not fully meet PLOS ONE’s publication criteria as it currently stands. Therefore, we invite you to submit a revised version of the manuscript that addresses the points raised during the review process.

We sincerely appreciate the substantial improvements you have made and the way you have addressed the previous reviewer comments.

Most of the concerns raised in the earlier round of review have been satisfactorily resolved. However, Reviewer 1 has now provided a few remaining theoretical concerns that require your attention before the manuscript can proceed further.

**Please consider the following points raised by Reviewer 1:**

**Introduction:**At the end of the discussion, you mention that your findings are consistent with online cognitive training platforms such as Lumosity. However, this is the first instance where Lumosity is mentioned in the manuscript. Since both your intervention and Lumosity report similar cognitive benefits, it is recommended that you introduce Lumosity earlier in the **Introduction** , to better highlight the novelty and relevance of your study.**Methods:**While you have described how the DATA and TECHS components are used in the intervention, their initial presentation remains somewhat abstract and contains redundancies. Given that these components are central to your cognitive training program, we encourage you to revise this section for improved clarity and organization. A clearer and more concise explanation will strengthen the reader’s understanding of your intervention model.

We kindly ask that you revise the manuscript in accordance with these comments and submit a revised version for further consideration.

Thank you once again for your efforts and your contribution to the field.

We look forward to receiving your revised manuscript.

Kind regards,

Roya Khanmohammadi, Ph.D

Academic Editor

PLOS ONE

Journal Requirements:

Reviewers' comments:

The author’s have presented a highly relevant cognitive stimulation program (MTT24.5), displaying improved cognitive abilities in healthy adults. Many of the previous concerns of the manuscript were resolved, however there are a few minor theoretical concerns, which lead me to my recommendation of “revise and resubmit”.

Introduction:

At the end of the discussion, the authors mention that their findings are consistent with online cognitive training platforms such as Lumosity. However, this is the first time the authors mention this online cognitive training platform, only at the end of the paper. Considering both Luminosity and the current study highlight similar benefits of enhanced cognitive abilities, it would strengthen the introduction and display novelty to the current study if Lumosity were introduced before the current study.

Methods:

The authors provided information on how the DATA and TECHS components are used for the intervention; however, the overall organization of these concepts throughout the methods section is not optimal. Especially when initially introducing the components, they are abstract and contain numerous redundancies. As these two components are central to the intervention, it is crucial that they are explained more clearly.

---

## [Author Response · Author response to Decision Letter 3]

3 Jul 2025

May 18, 2025.

Response letter to Academic Editor and to Reviewers

PONE-D-24-57989: A New Program for Systematically Enhancing Cognitive Reserve in Healthy Adults: A Pilot Randomized Active-Controlled Study

Dear Dr. Khanmohammadi, Ph.D, Academic Editor

Dear Reviewers, #1 and #2

We would like to express our sincere gratitude for the time and effort you have dedicated to reviewing our manuscript. We greatly appreciate your thoughtful feedback. In response to your comments, we have made the necessary revisions to the manuscript, as detailed below.

The two changes requested by Reviewer #1 were implemented, including the mention of the study on the cognitive platform Lumosity in the Introduction, and a rewritten description of the intervention in the Methods section to enhance clarity. Due to the inclusion of the Hardy et al. study in the Introduction, its position in the reference list was updated, which required adjusting the overall numbering of references accordingly.

These revisions address the points raised and we believe they have further strengthened the manuscript. We are hopeful that this second revised version meets your expectations and look forward to your feedback.

Thank you once again for your valuable support.

Sincerely,

Carol Kotliar, PhD

Responses to Reviewer #1

1. Comment (Introduction): “At the end of the discussion, the authors mention that their findings are consistent with online cognitive training platforms such as Lumosity. However, this is the first time the authors mention this online cognitive training platform, only at the end of the paper. Considering both Luminosity and the current study highlight similar benefits of enhanced cognitive abilities, it would strengthen the introduction and display novelty to the current study if Lumosity were introduced before the current study.”

Response: Thank you for your valuable observation regarding the mention of Lumosity. We agree that introducing this online cognitive training platform earlier in the manuscript improves the context and helps emphasize the novelty of our study. Following your recommendation, we have now included a mention of Lumosity in the Introduction section (page 3, lines 63 to 69).

2. Comment (Methods):” The authors provided information on how the DATA and TECHS components are used for the intervention; however, the overall organization of these concepts throughout the methods section is not optimal. Especially when initially introducing the components, they are abstract and contain numerous redundancies. As these two components are central to the intervention, it is crucial that they are explained more clearly.”

Response: We appreciate your comment regarding the organization and clarity of the DATA and TECHS components in the Methods section. We understand the importance of providing a clear and concise explanation, especially as these two components are central to the intervention. In response to your requirement, we have revised the description to improve clarity, eliminating abstract terms and redundant sentences. We have carefully reviewed the revised text to ensure its clarity and coherence. Please refer to pages 9 to 11 for the updated description.

---

## [Decision Letter · Decision Letter 3]

27 Jul 2025

Dear Dr. Kotliar,

Thank you for submitting your manuscript to PLOS ONE. After careful consideration, we feel that it has merit but does not fully meet PLOS ONE’s publication criteria as it currently stands. Therefore, we invite you to submit a revised version of the manuscript that addresses the points raised during the review process.

The statistical reviewer found your study to be reasonably well-designed, with appropriate data analysis. However, several minor but important issues were raised that require your attention before further consideration:

**Sample size justification** – Please clarify the rationale for using an effect size of 0.80. Was this based on prior literature, previous data, or another source?**Multiple comparisons** – Table 2 includes several secondary cognitive outcomes. Please indicate whether any corrections for multiple comparisons were applied, particularly for measures such as attention, verbal fluency, and visuospatial skills. Even in a pilot study, this should be addressed.**Randomization ratio** – While the 2:1 randomization appears justifiable, kindly explain the rationale for this choice explicitly in the manuscript to inform the reader.

We look forward to receiving your revised manuscript.

Kind regards,

Roya Khanmohammadi, Ph.D

Academic Editor

PLOS ONE

Journal Requirements:

Reviewers' comments:

Reviewer's Responses to Questions

**Comments to the Author**

Reviewer #3: (No Response)

2. Is the manuscript technically sound, and do the data support the conclusions?

Reviewer #3: Yes

3. Has the statistical analysis been performed appropriately and rigorously?

Reviewer #3: Yes

4. Have the authors made all data underlying the findings in their manuscript fully available?

Reviewer #3: Yes

5. Is the manuscript presented in an intelligible fashion and written in standard English?

Reviewer #3: Yes

Reviewer #3: The study is reasonably designed and the data analysis appropriate. However, there are several minor issues that should be considered.

1. One needs the rationale for the ES =0.80 in the sample size justification. Is it from past data, the literature or otherwise. Please explain.

2. There is no mention of multiple comparisons in the text or supplement protocol information. Table 2 has several secondary endpoints reported. Any adjustment of the p-values for multiple comparisons especially for Attention cognitive ability, Verbal fluency cognitive ability and Visuospatial skills cognitive ability. Even in a pilot setting this should be considered.

3. The need for the 2:1 randomization seems intuitively reasonable, but please explain the rationale for such for the reader.

**Do you want your identity to be public for this peer review?** For information about this choice, including consent withdrawal, please see our Privacy Policy

Reviewer #3: No

---

## [Author Response · Author response to Decision Letter 4]

5 Aug 2025

Dear Reviewer, #3

In response to your comments, we have made the necessary revisions to the manuscript, as detailed below.

The 3 minor revisions requested were implemented, including their detailed description for the readers inside the manuscript.

These revisions address the points raised and we believe they have further strengthened the manuscript. We are hopeful that this third revised version meets your expectations and look forward to the final acceptance of the manuscript.

Thank you once again for your valuable support.

Sincerely,

Carol Kotliar, PhD

Responses to Reviewer #3

1. Comment: “Sample size justification – Please clarify the rationale for using an effect size of 0.80. Was this based on prior literature, previous data, or another source?

Response: The effect size of 0.80 used in our sample size calculation was based on prior literature, particularly the methodology and effect size interpretation used in the ACTIVE trial (Advanced Cognitive Training for Independent and Vital Elderly).

Although the observed effects in ACTIVE ranged from small to moderate (e.g., 0.23 for reasoning and 0.66 for speed), we considered an effect size of 0.80 as a conservative target to detect a meaningful difference with sufficient power, assuming a best-case scenario for the intervention's impact.

Additionally, we considered the characteristics of our primary cognitive outcome measure. For example, in the Addenbrooke’s Cognitive Examination (ACE), a 1-point difference is clinically relevant for identifying cognitive decline. Therefore, assuming an intervention could achieve at least a moderate to large improvement (corresponding to 4 points, SD = 5.4, d = 0.8) was both clinically reasonable and consistent with previous literature on cognitive training effects.

We introduced this explanation also in the manuscript in the corresponding section to inform the reader. (Page 13)

2. Comment: “Multiple comparisons” Table 2 includes several secondary cognitive outcomes. Please indicate whether any corrections for multiple comparisons were applied, particularly for measures such as attention, verbal fluency, and visuospatial skills. Even in a pilot study, this should be addressed.

Response: Thank you for this important comment. We fully agree that the issue of multiple comparisons should be addressed, even in a pilot study. Given the exploratory nature of this pilot study, no corrections for multiple comparisons were applied. This decision was made to maximize sensitivity to potential signals that could inform future confirmatory studies. In pilot or exploratory studies like this one, it’s common not to apply these corrections. In these kinds of studies, the focus is more on detecting potential signals of an effect, even if that means accepting a higher risk of false positives.

Following your suggestion, we addressed it with a clearly state of this approach in the methods section as follows:

Changes in the manuscript:

A) We have added the following sentence to the Statistical Analysis section (at page 13):

"Given the exploratory nature of this pilot study, no correction for multiple comparisons was applied to secondary cognitive outcomes. This decision was based on the aim of generating hypotheses rather than confirming definitive effects. We acknowledge that this approach increases the risk of type I error, and therefore, the findings should be interpreted with caution and considered hypothesis-generating.”

B) We have also added a corresponding footnote in the manuscript at Table 2 (PAGE 14)

"P-values are unadjusted for multiple comparisons due to the exploratory nature of the study."

3. Comment: “Randomization ratio – While the 2:1 randomization appears justifiable, kindly explain the rationale for this choice explicitly in the manuscript to inform the reader.”

Response: A 2:1 randomization ratio was choice to support recruitment and enhance adherence among healthy volunteers by increasing the likelihood of being assigned to the intervention group. In studies involving cognitive enhancement in healthy populations, such an approach can be particularly effective in motivating participation. Given the nature of this pilot study, a larger sample size in the intervention arm allowed for a more informative preliminary assessment of feasibility and potential effects. This decision was made after careful consideration of ethical principles, study objectives, and available resources, ensuring a scientifically valid comparison between groups.

# See at manuscript PAGE 6: We incorporated the explanation as you requested into the manuscript to inform the reader.

Best regards,

Carol Kotliar, PhD.

---

## [Decision Letter · Decision Letter 4]

13 Aug 2025

A New Program for Systematically Enhancing Cognitive Reserve in Healthy Adults: A Pilot Randomized Active-Controlled Trial

PONE-D-24-57989R4

Dear Dr. Kotliar,

We’re pleased to inform you that your manuscript has been judged scientifically suitable for publication and will be formally accepted for publication once it meets all outstanding technical requirements.

Kind regards,

Roya Khanmohammadi, Ph.D

Academic Editor

PLOS ONE

Additional Editor Comments (optional):

Reviewers' comments:

Reviewer's Responses to Questions

**Comments to the Author**

Reviewer #3: All comments have been addressed

2. Is the manuscript technically sound, and do the data support the conclusions?

Reviewer #3: (No Response)

3. Has the statistical analysis been performed appropriately and rigorously?

Reviewer #3: (No Response)

4. Have the authors made all data underlying the findings in their manuscript fully available?

Reviewer #3: (No Response)

5. Is the manuscript presented in an intelligible fashion and written in standard English?

Reviewer #3: (No Response)

Reviewer #3: (No Response)

**Do you want your identity to be public for this peer review?** For information about this choice, including consent withdrawal, please see our Privacy Policy

Reviewer #3: No

---

## [Editor Report · Acceptance letter]

PONE-D-24-57989R4

PLOS ONE

Dear Dr. Kotliar,

I'm pleased to inform you that your manuscript has been deemed suitable for publication in PLOS ONE. Congratulations! Your manuscript is now being handed over to our production team.

Kind regards,

on behalf of

Dr. Roya Khanmohammadi

Academic Editor

PLOS ONE